# Climate is stronger than you think: Exploring functional planting and TRIAD zoning for increased forest resilience to extreme disturbances

Clément Hardy[1]*, Christian Messier[1,2], Yan Boulanger[3], Dominic Cyr[4], Élise Filotas[5]

**1** Department of Biological Sciences, Centre d'Étude de la Forêt (CEF), Université du Québec à Montréal, Canada (UQAM), Montréal, Québec, Canada, **2** Department of Natural Sciences, Institut des Sciences de la forêt tempérée (ISFORT), Université du Québec en Outaouais (UQO), Québec, Canada, **3** Natural Resources Canada, Canadian Forest Service, Laurentian Forestry Centre, Québec, Québec, Canada, **4** Sciences et Technology Branch, Environment and Climate Change Canada, Gatineau, Québec, Canada, **5** Department of Science and Technology, Université du Québec (TELUQ), Québec, Canada

\* hardy.clement@courrier.uqam.ca

## Abstract

In the face of global changes, forest management must now consider adapting forests to novel and uncertain conditions alongside objectives of conservation and production. In this perspective, we modified the TRIAD zoning approach to add a resilience component through functionally diverse plantations following harvesting in the extensive areas. We then assessed the capacity of this new "TRIAD+" zoning approach for improving the resilience of the mature forest biomass to climate change and three potential extreme pulse disturbances: a large fire, a severe drought, and an insect outbreak. We used the forest landscape simulation model LANDIS-II on a management unit in Mauricie (Quebec, Canada) to simulate and compare the TRIAD+ scenario with a classic TRIAD zoning scenario, and two business-as-usual harvesting scenarios with and without functional enrichment planting. We also simulated three different climate change scenarios (Baseline, RCP 4.5 and RCP 8.5) in which these management and extreme disturbance scenarios took place. We monitored the changes in three variables: the mature wood biomass across the landscape, the mature biomass of each functional group, and the functional diversity of stands in the landscape. Resilience was measured according to three indicators: resistance, net change and recovery time of mature biomass. TRIAD+ management resulted in a good compromise, harvesting the same amount of wood as other scenarios while increasing the surface of protected forests by around 240% compared to BAU scenarios, and improving the mean functional diversity of stands by around 15% compared to the classic TRIAD and BAU without plantations. Following the pulse disturbance events, TRIAD+ also increased the resilience of the mature biomass across the landscape. However, this increase was limited, depended on the resilience

**Data availability statement:** The data concerning the preliminary simulations to calibrate certain parameters of LANDIS-II, all of the parameter files for all of the simulations, the scripts used to launch the simulations on Compute Canada's clusters, the raw results and the scripts used to analyze the results and produce the figures are all available on the following Figshare repository: https://doi.org/10.6084/m9.figshare.25917739.

**Funding:** Funding was provided by the Natural Sciences and Engineering Research Council of Canada (NSERC) through a Collaborative Research and Development grant awarded to C. Messier (RDCPJ: 498998-16) with the joint support of Resolute Forest Products, and a Discovery grant awarded to E.Filotas (RGPIN: 2018-06156).

**Competing interests:** The authors have no competing interests to declare that are relevant to the content of this article.

indicator and the event considered, and was negligible in terms of tree biomass recovered in the long term. It's uncertain whether these results stemmed from the relative lack of small-scale interactions in LANDIS-II through which the effect of functional diversity on stand resilience should occur, or if this effect is small to begin with. Overall, our study reveals that an adaptation component can be included in current or future management strategies, but that increasing functional diversity via plantations will likely be insufficient to significantly boost forest resilience. Future research should therefore explore other (combined) means of increasing forest resilience, and improve the representation of small-scale interactions in landscape-scale models.

## 1. Introduction

Forests occupy a special place among the many components of the earth's biosphere. Holding much of the terrestrial biodiversity, they are crucial to the processes of life on Earth [1]. Forests also regulate the Earth's climate through their influence on the biogeochemical cycles of water, carbon and nitrogen, thereby playing an important role in climate regulation [2–4]. For humans, forests also represent places of wonder and spirituality, homes, and vital sources of different resources, ranging from construction material to food or medicine [1,5]. Yet these numerous roles played by forests are now under threat from many different environmental pressures [6–8]. Chief among them is global change – the combination of direct anthropic pressures, human-induced climate change, and other processes interacting with humans such as the invasion of exotic species and pathogens [9–11].

Forestry is one of the pressures that global change applies on forests. While humanity has been harvesting wood from time immemorial, the quantity of wood harvested throughout the world has grown enormously since the Industrial Revolution. In recent years, this quantity has surged from 2.5 billion m³ of roundwood per year produced worldwide in 1960 to almost 4 billion m³ in 2020 [12,13] through increasing human demographics and the emergence of new harvesting technologies that have made harvesting cheaper and faster [14]. Wood and timber production worldwide has shown no sign of slowing but is instead increasing yearly. Forestry is thus a recurrent and intense disturbance affecting forests around the world, leading to their gradual transformation. This transformation occurs through practices such as the selection of species of commercial interest [15], or through the large-scale application of methods such as clear-cutting that reduces the quantity of older forests [16]. Forestry has thus changed the structural diversity [17,18], species composition [19], and connectivity [20] of forests at large temporal and spatial scales. Consequently, forestry has also altered the habitat that forests provide to many other species and tempered their ability to sustain important ecological functions [21].

In contrast to these trends, several authors have recently proposed a more optimistic view of forestry. In this view, the inherent capacity of forestry to influence the structure and composition of forests could be used as an advantage. Specifically, forestry could be used as an opportunity to diversify forests across different spatial

scales to improve their resilience to the uncertain future perturbations caused – or influenced – by global changes [22–24]. In this way, forestry would join the global effort to increase the diversity of responses that both human and natural systems exhibit in the face of disturbances [25,26]. A potential way to bring ecosystems to a state of higher resilience is by using the concepts of functional response traits and response diversity. Functional response traits aim to capture the biological, structural or behavioral characteristics of an individual associated with its response to environmental changes (e.g., root length, bark thickness, etc.; [27]) and its effects on the environment. As such, a community presents a range of responses to the environment among its organisms (e.g., individuals, species) depending on the variability between their functional response traits [28]. Therefore, it is suggested that a community with a high response diversity will be more resilient to disturbances, as it increases the chance that some individuals have the favorable traits to resist or recover [29–32]. Forests could thus be diversified or altered by forest management practices at both stand and landscape levels to present a higher functional response diversity to future disturbances, and, as a result, a greater resilience [33–35].

While this new vision of forestry based on preparing forests for future conditions is promising, it differs from current forest management strategies in several ways. The main difference relates to the prevailing management goal of conserving or retrieving a "reference state" of forests. For example, strategies adopted in many temperate and boreal forests promote an ecosystem-based forest management [36] or close-to-nature forestry [37,38], which focus on emulating the "historical range of variation" of natural disturbances through forest management. According to this approach, forest cuts should reproduce the size, intensity and impacts of natural disturbances with which forests have evolved and developed regeneration mechanisms. Yet, applying this approach is complex due to the difficulty of obtaining reliable historical characteristics of natural disturbances. Moreover, recent studies have criticized the implementation of ecosystem-based management that focus exclusively on reproducing past disturbance patterns as current and future forests are likely to be confronted with novel environmental conditions and disturbance regimes [31,39,40]. Hence, adding forest resilience to global change as a new management goal may be at odds with current management strategies that are largely focused on wood production and conservation.

A potential approach for addressing this challenge is to amend current management strategies by integrating the notion of resilience, adaptation and guided change of forest structure and composition. In this article, we propose a resilience-based modification of the TRIAD zoning approach, originally developed by Seymour and Hunter [41]. The conventional TRIAD approach consists of dividing a managed forest landscape into different specialized areas dedicated to different goals: intensively managed areas to maximize production, extensively managed areas to accommodate a broader range of ecosystem services and conservation objectives, and reserves for conservation purposes only [42]. These zones are positioned in the landscape in order to minimize trade-offs between conservation, production, and social acceptability (e.g., setting intensive areas far from conservation areas) while supplying the same amount of harvested wood as a landscape without zoning [43]. Intensive zones aim at maximizing productivity such that harvest targets are reached more easily on smaller areas, thus allowing the size of conservation areas to be increased [44]. Because of these potential benefits, the TRIAD zoning is currently being tested in several areas of the world [44,45].

The TRIAD approach offers the opportunity to accommodate the new resilience objective by increasing the functional response diversity of forest stands in extensively managed zones. This can be done through enrichment planting, which consists of planting trees in an already existing forest overstory that has been thinned [46]. It is further called functional enrichment or functional planting when used to diversify the functional response traits present in the forest rather than simply improving species richness [34]. Therefore, functional enrichment of forests could improve the response to future disturbances as the forests would contain a more diversified portfolio of response traits [29,47]. Through long-term planning, the establishment of functionally enriched plantations distributed across the landscape could allow resilience to scale up from the plantation to the landscape scale through seed dispersal [31,48]. We call this TRIAD +, a new version of TRIAD that includes functional enrichment via plantations.

While this avenue is appealing, several factors must be investigated to assess the efficacy of functional enrichment plantations – and the TRIAD+ as a whole – to improve forest resilience. For example, future climatic conditions could make it difficult for the species selected for functional diversification to coexist at the stand scale if their growth is impeded by changes in temperature or water availability. Also, the accumulation of more frequent, severe and varied disturbances might increase tree mortality or hamper their growth even in functionally rich plantations (e.g., through repeated fires, droughts, windthrow, etc.; [49]). Functional enrichment may also require productive sites to successfully plant and grow certain species, thereby making these sites unavailable for intensive management areas. Finally, implementing functional enrichment plantations will often require preliminary forest cuts that, if too numerous in the landscape, might negatively impact forests in other ways, canceling the positive effects of such enrichment. For example, such cuts might change the age distribution towards younger forests in the landscape [16], necessitate more forest roads [50], and degrade the habitat quality for certain wildlife specialist species [21,51].

In this study, we set out to explore these uncertainties by measuring the potential of functional enrichment planting to improve the resilience of a vast forest landscape to different disturbances. Specifically, we assessed the capacity of a TRIAD+ zoning approach to improve the resilience of the mature forest biomass to climate change and three future potential extreme disturbance events: a large fire, a severe drought, and an insect outbreak. Using the forest landscape simulation model LANDIS-II [52] on a management unit in Mauricie (Quebec, Canada), we simulated and compared the TRIAD+ scenario with a classic TRIAD zoning scenario, and two business-as-usual harvesting scenarios with and without functional enrichment planting. Furthermore, we simulated three different climate change scenarios (Baseline, RCP 4.5 and RCP 8.5) in which these management and extreme disturbance scenarios took place. We then measured the resilience of the mature forest biomass, at the landscape scale, following one of the three extreme disturbance events (fire, drought or insect outbreak). Finally, we assessed the benefits of each management scenario by considering the trade-offs between the objectives of production, conservation and adaptation.

## 2. Methods

### 2.1. Simulated area

Our simulated area is a forest landscape extending over more than 4 million hectares in the Mauricie region (Quebec, Canada; Fig 1). It consists of a forest management unit surrounded by a 50 km buffer zone, which has been simulated with LANDIS-II in a previous study [50]. As of 2020, the landscape comprised 339 117 ha of protected forests or around 9% of the total forest surface of the simulated area. These protected forests were divided into approximately 400 areas, five of them being relatively large (> 20 000 ha) and most of them relatively small (< 300 ha). The southern area is mainly composed of mixedwood forests dominated by balsam fir (*Abies balsamea*), yellow birch (*Betula alleghaniensis*) and trembling aspen (*Populus tremuloides*). In contrast, the northern area is a boreal coniferous forest dominated by balsam fir, white birch (*Betula papyrifera*), trembling aspen, black spruce (*Picea mariana*) and jack pine (*Pinus banksiana*). Forest fires are an important disturbance in the north, while spruce budworm (*Choristoneura fumiferana*) outbreaks are present in the south [53,54]. Being located at the transition from the temperate to the boreal forest, this study area thus presents a clear dichotomy of forest composition and natural disturbances, making it a good choice for exploring the effects of functional enrichment and zoning in two contrasting ecological contexts.

### 2.2. Experimental design

To explore the effects of functional enrichment and zoning on forest resilience, we simulated forest dynamics over 200 years using different scenarios that varied three distinct factors: the forest management strategy used, the intensity of climate change, and the occurrence of a catastrophic disturbance event imposed at year 100 of the simulation (Fig 2). The precise implementation of each of these factors within LANDIS-II is detailed in the following sections.

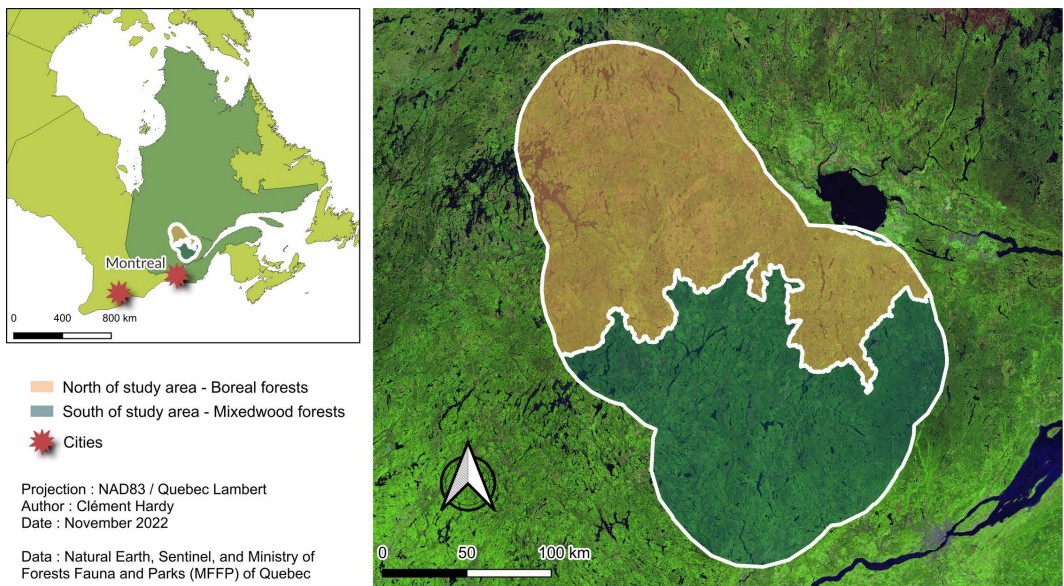

**Fig 1. Map showing the location and extent of our study area located in the Mauricie region of Quebec, Canada.** Satellite data from the Sentinel satellite, and edited by the Ministry of Forests and Natural Resources of Quebec under a CC-BY 4.0 license (https://www.donneesquebec.ca/recherche/dataset/mosaique-satellites).

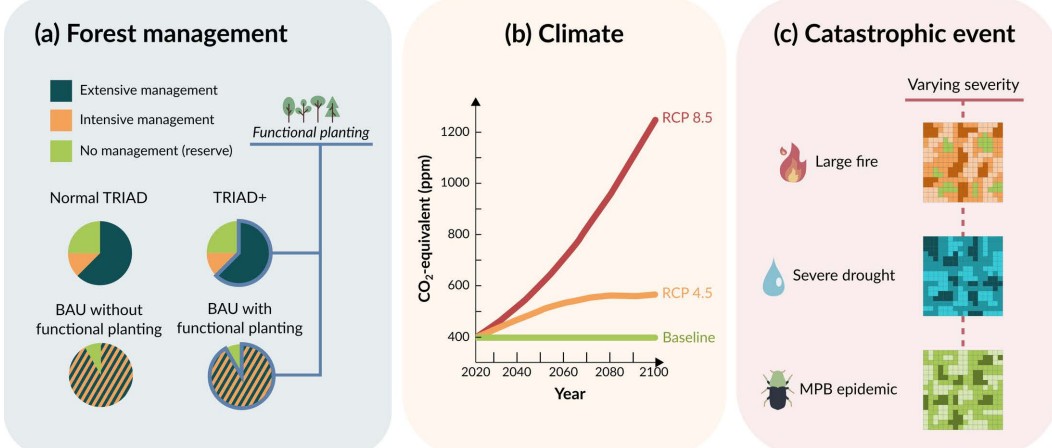

**Fig 2. Visual representation of the three categories of scenario simulated with LANDIS-II: forest management, climate, and catastrophic event.**

We designed four forest management strategies, each defined by the presence or absence of functional enrichment through planting (hereinafter referred to as "functional planting") and of TRIAD zoning (Fig 2a). The first two strategies were a "Business as Usual" (BAU) scenario with (BAU-PlantFunct) and without (BAU-NoPlant) functional planting. In these scenarios, no TRIAD zoning took place: forest harvesting of different intensities could be carried out anywhere in the landscape except in the current protected areas. All simulated prescriptions are detailed in "Harvesting". In essence, the BAU-NoPlant scenario acted as a control scenario for our study area. The other two strategies were TRIAD+, with

functional planting, and the "normal" TRIAD, without functional planting. In these scenarios, the landscape was divided into extensive, intensive, and protected areas, with functional planting taking place in the extensive areas of TRIAD+ and intensive areas located on the most productive forests of the landscape (see "Harvesting"). The existing protected areas were expanded using a 5000 m buffer compared to the BAU scenarios, increasing the total percentage from around 9% to 24% of the forested area. This allotment was devised to match the philosophy of TRIAD zoning where intensive areas are made as productive as possible to spare more forest from exploitation [42,44]. Importantly, all four management scenarios (BAU-PlantFunct, BAU-NoPlant, TRIAD, TRIAD+), had to harvest the same amount of biomass at every 10-year time step. This biomass target was based on the current harvest levels in the simulated area during the period 2018–2023, as indicated in documents produced by the Ministère des Forêts, de la Faune et des Parcs du Québec [50,55].

The climate varied according to three different Representative Concentration Pathway (RCP) scenarios [56]: Baseline (no change in climate as compared to 2020) which served as a control, RCP 4.5, and RCP 8.5 (Fig 2b). Three catastrophic disturbances were chosen to represent potential, and quite unpredictable, future disturbance events that could take place in our study area, triggered by the effects of global change. Our disturbance scenarios consisted of a large forest fire covering 70% of our simulated area; an intense drought; and an outbreak from an insect not yet present in the area (Fig 2c). We also simulated scenarios without the occurrence of these catastrophic disturbances to serve as a control. Fire and drought were chosen as they are both expected to increase in frequency and severity with climate change, making the occurrence of extreme cases more probable [11,57]. In Quebec, a single fire of more than 1.2 million hectares was indeed observed in 2023 [58] during a fire season that resulted in more than 5 million hectares of forest burned at the provincial scale [59]. This fire season was caused, in part, by drought [60], which Global Climate Models predict will become more frequent and intense in the future [61]. Such large fires could become even larger in the future, according to observed trends in Canada [62]. For the insect disturbance, we selected the Mountain Pine Beetle (*Dendroctonus ponderosae;* MPB), a wood-boring species of bark beetles whose ongoing outbreak in western Canada is causing excessive damage by attacking a wide range of pine species. Previous studies have shown the possible future expansion of the MPB into the forest of eastern Canada, helped by a changing climate and by the presence of host species [63–65]. Therefore, we defined a scenario where a large outbreak of MPB would impact the pine species of our landscape, to which forest managers would be initially unprepared. Simulating this disturbance allowed us to study a potential undesired effect of functional plantations, as increasing pine trees in the landscape could increase its sensitivity to MPB outbreaks.

We simulated one scenario for each unique combination of these three factors (management, climate, and catastrophic disturbances), resulting in 48 distinct scenarios. We accounted for stochasticity associated with wildfires, seed dispersal and regeneration by running five replicates for each scenario, resulting in an ensemble of 240 individual simulations. In each simulation, we measured two variables of interest at the landscape scale (total mature biomass and mean functional response diversity) and we assessed the variations between these measures according to the three different factors (management, climate, catastrophic disturbances). We also measured three indicators of resilience to the catastrophic disturbances (see "Data analysis") to evaluate whether the different forest management strategies were associated with increased or decreased forest resilience values. Moreover, we assessed whether an increased resilience implied a trade-off with another variable of interest (e.g., mature biomass).

## 2.3. LANDIS-II model

LANDIS-II is a spatially explicit Forest Landscape Model (FLM) that simulates forest dynamics via two main processes: forest succession (growth, mortality, recruitment, etc.) and natural or human-induced forest disturbances (harvesting, forest fire, insect outbreaks, etc.) [52]. The processes are individually simulated by extensions that are activated sequentially during each time step. These extensions are chosen by the user and can simulate the dynamics of different ecological variables (e.g., biomass, carbon stocks, etc.). In LANDIS-II the simulated landscape is composed of square cells

representing a forested or non-forested area (e.g., water, urban area, etc.). All forested cells are assigned to different ecoregions to integrate the effect of climate and soil into the simulated processes.

**2.3.A. Core parameters.** The main parameter values used in our study were derived from the protocol of several recent studies that used LANDIS-II to simulate forest landscapes in Quebec [66,67]. We simulated 17 different tree species that were among the most abundant in our study area, with their life-history traits being derived from several sources (books and previous studies; see S1 Appendix A in S1 File and [68]). We used a grid size of 100 m (1 ha) and a time step of 10 years as a compromise between computation time and level of detail, as is often done in LANDIS-II studies [68–70]. Finally, we set the total simulation length to 200 years in order for the simulated forest management strategies to take effect in the landscape during the first 100 years, and then measured the response of stands impacted by the catastrophic disturbance events happening at t = 100 during the remaining 100 years.

The initial composition and structure of the forest in each grid cell were established using ecoforestry maps from the province and data from cohort studies conducted in the province's permanent and temporary forest inventory plots [55,71,72]. The composition and age structure of the inventory plots were translated into LANDIS-II age-cohorts. These age cohorts were then assigned to the forest stands identified in the ecoforestry maps through the k-NN method [68]. Furthermore, we defined forest stands as groups of forested cells having identical composition, age structure and abiotic conditions according to the 5th provincial forest inventory of Quebec [73]. The position of forest stands remained constant through time.

These maps were converted into a raster format with a resolution of 250 meters (equivalent to 6.25 hectares). Subsequently, each cell was allocated to a uniform spatial unit, known as a "land type," characterized by consistent soil and climatic conditions [67]. Cells with over half of the area occupied by non-forest cover types were categorized as non-active.

**2.3.B. Biomass succession.** We used the Biomass Succession extension (v5.2) of LANDIS-II to simulate the succession dynamic and the living aboveground tree biomass of age cohorts within forest cells. This extension uses three important parameters that vary for each species, ecoregion, and time step: the probability of establishment, the maximum growth rate, and the maximum biomass that an age cohort can reach. These parameters were obtained using PICUS [74], an individual-based model that simulates tree growth at the stand scale for specific soil and climatic conditions. Following the methodology of Boulanger and Pascual Puigdevall [67], we used projections of future climate data from the Canadian Earth System Model version 2 (CanESM2) and soil data from Sylvain et al. [75] in PICUS simulations. From these simulations, we derived species-specific parameters for each climate scenario (baseline for 2020, RCP 4.5 and RCP 8.5) and for all ecoregions of the landscape. The available climate projections only extend to 2100. Since our investigated scenarios were simulated until 2220, we assumed that climate conditions beyond 2100 remained constant.

The other parameters required by the Biomass Succession extension – such as growth curves or the impact of shade on productivity – were derived from calibration runs of LANDIS-II. The goal of these calibrations was to reduce the difference between the initial estimates of biomass by the extension, and the biomass estimates from remote sensing [76]. These biomass estimates computed by Biomass Succession are based on the initial community structure for each cell (see "Core parameters") following the methodology described in [77].

**2.3.C. Base fire.** Throughout all simulations, we simulated the natural disturbance regime specific to this landscape: forest fires and spruce budworm outbreaks. The catastrophic pulse disturbances (see "Catastrophic disturbance events") therefore occur in addition to these natural recurrent disturbances. Forest fires were simulated with the Base Fire extension (v4.0) [78]. This extension simulates fire ignition and propagation in the landscape based on three characteristics that together determine a fire regime: fire size, number of fires and fire severity. Different regions with specific fire regimes can be determined by the user. We defined two homogeneous fire regions in our landscape using the methodology of Boulanger et al. [68] and the data from Boulanger et al. [79]. For each of these two fire regions, we used calibration runs in LANDIS-II to find parameters for the Base Fire extension that would replicate the correct minimum and maximum size of fires and the percentage of annual area burned. As the fire regime in each region changes with

time due to the shifting climate, we obtained a set of parameters for each simulated climate scenario, and for three time periods (2020–2040, 2041–2070, 2070-beyond) according to the predictions of Boulanger et al. [79]. The fire regions in the simulated landscape presented a high variability in fire size each year. To account for this stochasticity, we simulated 30 replicates of each calibration run to obtain an average annual area burned corresponding to the existing projection for every climate scenario [79]. We only simulated forest fires with the highest severity, corresponding to crown fires, as those are the most frequent and important fires in our study area [80].

**2.3.D. Spruce budworm.** We simulated spruce budworm (SBW) outbreaks via the Biological Disturbance Agent (BDA) extension (v4.0.1). The extension simulates new epicenters as probabilistic events in landscape cells, from which outbreaks propagate to surrounding cells. Cells are disturbed with different degrees of severity depending on their host proportion which in turn influences the probability of mortality of the different age cohorts. Host tree species for the SBW were, from most to least vulnerable, balsam fir (*Abies balsamea*), white spruce (*Picea glauca*), red spruce (*Picea rubens*) and black spruce (*Picea mariana*). While climate is predicted to alter outbreak dynamics by changing the tree species composition across the landscape [81], for simplicity we omitted any direct effects of climate change on SBW outbreaks. We parameterized the BDA extension using parameters from Boulanger et al. [82]. This study derived the parameters through a calibration and validation exercise using a forest ecosystem landscape similar to the one present in our study area. This parametrization led to the simulation of SBW outbreaks with a periodicity of 40 years and a duration of 10 years [54].

**2.3.E. Harvesting.** We developed a new harvesting extension for LANDIS-II named "Magic Harvest" that works in tandem with the existing harvest extensions of LANDIS-II [83]. This extension allowed the implementation of more complex harvest prescriptions at the stand level. Using this new extension, we implemented six different types of stand-level harvest prescriptions (Table 1). *CC-PlantIntens* consisted of a complete clearcut followed by the establishment of "intensive" plantations. These plantations combined a fast-growing hybrid tree species with a marketable tree species (e.g., black spruce) to maximize wood production. The model parameters used for characterizing the growth of these hybrid species were defined to emulate the rapid growth of hybrid poplar and hybrid larch recently developed in North America. As such, the main physiological parameters of these hybrid species (e.g., shade and fire tolerance) were similar to their non-hybrid alternatives (the trembling aspen and the tamarack; see [84] and [85]), but their maximum Annual Net Primary Productivity (ANPP) was doubled, and their maximum biomass was increased by 15% in every ecoregion. Moreover, the longevity of these hybrid species was divided by two, leading to the faster mortality. We based these parameter changes on expert opinion and on data from the Quebec Sylvicultural

**Table 1. Description of the harvest prescriptions simulated.**

|  | Cut | | Planting | Repetition |
|---|---|---|---|---|
|  | Percent biomass removed | Age of cohorts | Species planted | Timing of prescription |
| **CC-PlantIntens** | 100% |  | Hybrid poplar or hybrid larch depending on the latitude, combined with black spruce | None |
| **CC-PlantFunct** | 90% | > 10 | Species from functional groups absent or rare in the stand | None |
| **CC-NormalPlant** | 90% | > 10 | Dominant species in the stand | None |
| **CC-NoPlant** | 90% | > 10 | None | None |
| **Selection Cutting (SC)** | 30% | >= 30 | None | Every 30 years for 90 years |
| **Commercial Thinning (CT)** | 80%<br>66%<br>60%<br>40%<br>5% | <= 30<br>31-50<br>51-90<br>91-100<br>> 120 | None | • At year 20 and year 50 in TRIAD intensive zones;<br>• No repetition in BAU. |

Guide [86]. *CC-PlantFunct, CC-NormalPlant* and *CC-NoPlant*, all consisted of a clearcut with protection of advanced regeneration; in *CC-PlantFunct* the clearcut was followed by the establishment of functionally enriched plantations, in *CC-NormalPlant* the clearcut was followed by tree planting of the dominant species prior to harvesting, and in *CC-NoPlant* the clearcut was not followed by any kind of plantations. *SC* consisted of a selection cutting, harvesting 30% of the stand biomass of all age cohorts older than 30 years, every 30 years during a 90-year period. Finally, *CT* consisted of a commercial thinning, removing 60–80% of the biomass of younger tree cohorts but only 5–40% of the biomass of older cohorts and was repeated twice in a 50-year interval in TRIAD intensive areas. Stands restricted for harvesting with SC and CT became available to any other prescription after the 90- (*SC*) or the 50-year (*CT* in intensive TRIAD areas) period. The details of each prescription are given in S1 Appendix B in S1 File. All prescriptions targeted the stands with highest biomass available for harvest in the landscape or in their area of application (see below), except for *CC-PlantFunct* which targeted the stands with the lowest functional response diversity (see "Data analysis"). Prescriptions were applied in an arbitrary order, harvesting their biomass target one after the other until all prescriptions for the given scenario were considered.

Furthermore, in the enriched plantations of *CC-PlantFunct*, species were selected based on the functional groups present in the targeted stand. The functional groups in this study were identified through a clustering analysis of the 17 simulated species using nine different functional response traits related to our three catastrophic disturbances (see "Catastrophic disturbance events"): maximum height, seed dry mass, wood density, leaf nitrogen content per leaf dry mass, specific leaf area (SLA), bark thickness, fire tolerance, drought tolerance and shade tolerance (S1 Appendix D in S1 File). The clustering method resulted in five functional groups: three gymnosperm groups with different tolerances to shade, drought, and fire; and two angiosperm groups with a clear distinction between pioneer and mid to late-succession species. When species from one or several functional groups were not present in the targeted stand, we planted one new age cohort of one species for each missing functional group. The selected species for each missing group was chosen randomly among those with the highest probability of establishment, which varied by ecoregion and could thus differ with time due to climate change. If all functional groups were already present in the stand, we selected a species from the group with the smallest abundance (as estimated by their biomass). This methodology ensured a local increase in functional response diversity.

The four forest management strategies harvested the same biomass target in different ways (Table 2). In the two types of BAU scenarios, all unprotected forests were available for harvesting by *SC*, *CT*, and *CC-NormalPlant*. In addition, *CC-PlantFunct* was available in BAU-PlantFunct and *CC-NoPlant* in BAU-NoPlant. In contrast, the TRIAD+ and normal TRIAD scenarios restricted the use of *SC*, *CC-PlantFunct* (in TRIAD+) and *CC-NoPlant* (in Normal TRIAD) to their extensive zones. In the intensive zones, forests were instead harvested with *CT* and *CC-PlantIntens* to maximize wood production using hybrid species and commercial thinning. As such, thinning (*CT*) in the intensive areas of TRIAD scenarios was repeated twice during the 50-year period following a first thinning to simulate an intensive commercial thinning (Table 1).

**Table 2. Percentage of the biomass target harvested with the different harvest prescriptions in the four forest management scenarios.**

|  | CC-PlantIntens | CC-PlantFunct | CC-NormalPlant | CC-NoPlant | SC | CT |
|---|---|---|---|---|---|---|
| **TRIAD+++** | 25% [I] | 37.5% [E] |  |  | 12.5% [E] | 25% [I] |
| **Normal TRIAD** | 25% [I] |  |  | 37.5% [E] | 12.5% [E] | 25% [I] |
| **BAU-PlantFunct** |  | 37.5% [A] | 45% [A] |  | 12.5% [A] | 5% [A] |
| **BAU-NoPlant** |  |  | 45% [A] | 37.5% [A] | 12.5% [A] | 5% [A] |

I : *In intensive areas only*

E : *In extensive areas only*

A : *in all the landscape except protected areas*

The intensive zones were fixed and corresponded to 16% of the forest surface with the highest Annual Net Primary Productivity (ANPP) during a calibration run made with no natural disturbances. The gain in productivity in the intensive zones allowed us to increase the size of the unharvested areas. By creating a 5000 m buffer around the largest current protected areas (as of 2020), conservation reached 24% of the forested area. Only the size of the 10% largest protected areas was increased. Most existing small, protected areas were created by the provincial government to act as biological refugia dispersed across the landscape [87,88]. In agreement with this original goal, the size of small, protected areas was not increased. Thus, the TRIAD zoning ratio consisted of 16% of the forested surface dedicated to intensive management, 60% to extensive management, and 24% to conservation. This ratio is similar to the one recommended by Blattert et al. [45] in their estimation of an optimal TRIAD zoning in Finnish landscapes.

**2.3.F. Catastrophic disturbance events.** We simulated catastrophic disturbance events in LANDIS-II using the Biomass Harvest extension where mortality is represented by the loss of biomass in impacted forest stands. While originally designed to simulate harvesting, this extension can be employed to simulate any disturbance by removing tree biomass according to different severity and spatial distribution patterns. The severity of these events at the stand scale, i.e., the amount of removed biomass of a given species age cohort, was determined based on the functional response traits of that species as well as those of the other species in the community. The severity of disturbances also varies according to small-scale factors such as topography, soil, and microclimate. For simplicity, we did not include these factors and rather focused on the influence of species composition on disturbance severity.

**Large fire:**  We defined the large fire as a disturbance extending across the entire landscape but creating numerous unburned forest patches, i.e., fire refugia (Fig 2c). The total area covered by the refugia was fixed at 30% of the landscape, a proportion intermediate to the minimum (20%) and maximum (57%) values determined by Walker et al. [89] when measuring fire refugia from satellite imagery in coniferous and mixed forest landscapes. The size of individual refugium was sampled from a power-law distribution varying between one and 100 ha to make large refugia uncommon [89,90] (S1 Appendix C in S1 File). Each refugium was created by first randomly choosing a forest stand in the landscape to be at the center of the refugia, and then increasing its size from stand to stand until the sampled size was reached. For simplicity, the stand at the center of the refugia was selected randomly since modelling the influence of fine scale factors (e.g., slope, topographic wetness, etc.) on the creation of refugia was beyond the scope of our study [91]. Refugia were added one by one in the landscape until 30% of the landscape's surface was reached.

Within burned stands, we assumed that tree biomass was consumed by fire according to a species-level process and a stand-level process. At the species level, we used the fire tolerance trait to determine the proportion of biomass loss of an age cohort. This proportion varied from a 60% loss at high tolerance (i.e., trait value between 4 and 5) to 100% at low tolerance (trait value less than 1) [78] (Table 3). At the stand level, we computed the Community Weighted Mean (CWM) of the fire tolerance trait over all species present (Table 3 and Fig 2c). This stand-level tolerance measure served to determine a protection effect from the community which in turn modulated the loss of biomass of each age cohort within the stand (based on [92]). The protection effect varied from 0 (no protection), when the stand CMW fire tolerance was less than 1, to 40% when it was highest (value between 4 and 5) (Table 3). The resulting biomass loss of an age cohort due to the large fire was calculated by multiplying the stand-level protection effect with the species-level proportion of biomass loss. These species and stand-level processes were parameterized based on our expert knowledge of forest fires and fire tolerance since the existing literature could not provide direct parameter values (Table 3).

Although biomass loss was a function of fire tolerance during this single catastrophic fire, periodic fires generated by the Base Fire extension consumed all biomass irrespective of species tolerance to fire. Indeed, the Base Fire extension was used to simulate smaller but more intense fires, whereas the Biomass Harvest extension was used to simulate an extremely large and long fire event varying in intensity according to forest composition. In addition, following the large fire, we did not simulate the regeneration of serotinous species, as their fire tolerance implied that their age cohorts would never entirely disappear from a burned cell.

**Table 3. Proportion of biomass loss for each catastrophic event at the species and stand level. The total proportion of biomass loss for a given species in a given stand is computed by multiplying both species age cohort and stand level effects.**

| Catastrophic event | Age-cohort level | | Stand level | |
|---|---|---|---|---|
| | **Factor influencing biomass loss** | **Biomass loss (%)** | **Factor influencing the reduction of biomass loss** | **Reduction of biomass loss (%)** |
| **Large fire** | **Species fire tolerance** | | **Stand CMW of fire tolerance** | |
| | 0-1 | 100 | 0-1 | 0 |
| | 1-2 | 90 | 1-2 | 10 |
| | 2-3 | 80 | 2-3 | 20 |
| | 3-4 | 70 | 3-4 | 30 |
| | 4-5 | 60 | 4-5 | 40 |
| **Severe drought** | **Species drought tolerance** | | **Stand functional diversity** | |
| | 0-1 | 70 | 0-1 | 0 |
| | 1-2 | 60 | 1-2 | 5 |
| | 2-3 | 50 | 2-3 | 10 |
| | 3-4 | 40 | 3-4 | 15 |
| | 4-5 | 30 | 4-5 | 20 |
| **MPB epidemic** | **Host status** | | **Stand host abundance (%)** | |
| | | | 100−90 | 0 |
| | *Pinus strobus* | 80 | 90−80 | 8.6 |
| | | | 80−70 | 17.2 |
| | *Pinus resinosa* | 80 | 70−60 | 25.8 |
| | | | 60−50 | 34.4 |
| | *Pinus banksiana* | 80 | 50−40 | 43 |
| | | | 40−30 | 51.6 |
| | Non-host species | 0 | 30−20 | 60.2 |
| | | | 20−10 | 68.8 |
| | | | 10−0 | 77.4 |

**Severe drought:** In contrast to the refugia created during the large fire disturbance event, the severe drought affected the entire landscape. The loss of biomass resulting from drought mortality was also determined through a species-level and a stand-level process. For each species' age cohort, biomass loss depended on the species' drought tolerance trait [93]. At the stand level, higher values of functional response diversity (see "Data analysis" for its measurement) increase the drought tolerance for all species present. This stand-level effect was based on studies suggesting that functional response diversity was more important than species diversity in improving drought tolerance through resource partitioning and facilitation (S1 Appendix C in S1 File; [94], [95]). As for the large fire, we parameterized this process based on expert estimation of the biomass lost for species of different drought tolerance and according to the stand functional diversity.

**Mountain pine beetle outbreak:** The MPB outbreak affected all forest stands containing any of the potential host species: white pine (*Pinus strobus*), red pine (*Pinus resinosa*), and jack pine (*Pinus banksiana*) (Table 3). We assumed that age cohorts of these species lost 80% of their biomass. We based this proportion on the study of Long and Lawrence [96] which reported a pine tree mortality exceeding 80% in MPB-infected landscapes of western Montana (USA). At the stand level, we hypothesized that a dilution effect from the presence of non-host species would reduce host mortality caused by MPB [97]. Therefore, we assigned a protection effect that increased as a function of the abundance of non-host species present in the stand. We used the study of Jactel et al. [98] measuring the effect of stand diversity on damages caused by borer insects to estimate this stand-level protection effect (see S1 Appendix C in S1 File, Table 3). Again,

the resulting loss in biomass for a given age cohort was calculated by multiplying the proportion of biomass loss at the species-level with the protection effect at the stand-level.

## 2.4. Data analysis

In each scenario, we measured two variables associated with forest ecosystem functions. Firstly, we measured the biomass of mature cohorts (defined as being 40 years or older) of all species present in each stand of the landscape. The mature biomass is an important proxy for several forest functions like carbon storage and seed production, as mature trees can reproduce and create seeds while storing more carbon than smaller trees [99,100]. In addition, mature trees are important to the forest industry given the higher market value of large diameter trees. The total mature biomass in the landscape ($B_L$) was calculated by summing the stand-scale mature biomass ($B_S$) across all stands. Secondly, we measured the functional response diversity in each stand ($FD_S$) using the exponent of the Shannon's diversity index, applied to the relative biomass abundance of each functional group in the stand:

$$FD_s = \exp\left(-\sum_{i=1}^{n} p_i \cdot \log(p_i)\right)$$

(1)

where $p_i$ is the relative abundance of functional group $i$ (expressed by the biomass of its age cohorts) from the $n$ functional groups present in stand $S$. $FD_S$ measures the effective number of functional groups in a stand, quantifying its functional response diversity in a simple yet meaningful way [35,101]. Furthermore, we computed the mean functional response diversity across stands of the landscape ($\overline{FD}_S$) as a stand-area weighted mean of $FD_S$. We used $\overline{FD}_S$ to observe if functional planting did effectively increase the functional response diversity of stands at the landscape scale, and if this increase could further be linked to changes in our resilience measures (see below). In addition, we computed the total biomass of all age cohorts in the landscape for each functional group (referred as $B_{FG}$). The three resulting measures ($B_L$, $\overline{FD}_S$ and $B_{FG}$) were subsequently averaged across the five simulation replicates for each scenario combination.

Moreover, we measured the resilience of the mature biomass at the landscape scale ($B_L$) following one of the catastrophic events at $t = 100$. Resilience is a notably complex concept to capture that has led to the development of varied measures (e.g., speed of recovery, critical slowing down, etc.) [102]. In the context of this study, we define resilience as the ability of a system to maintain essential functions in the face of a disturbance [33]. In particular, we followed the methodology of Cantarello et al. [103] and measured the resilience of $B_L$ through three different metrics (Fig 3): resistance (R), net change (NC) and rate of recovery (RR). As noted above, the mature biomass acts here as a proxy for several important ecosystem functions such as seed production, carbon storage, and wood production.

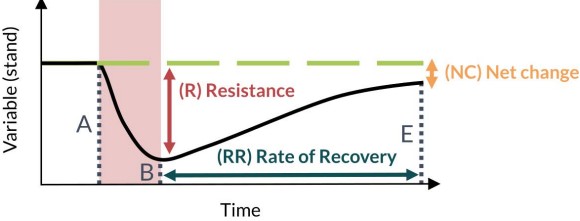

**Fig 3. Measures of resilience used in our study.** A, B and E refers to the value of the variable of interest before the disturbance (A), right after it (B), and at the end of the simulation (E), and are used in equations 2 and 3.

R was defined as the variation between the value of the variable immediately before ($B$, at $t = 90$) and after ($A$, at $t = 100$) the catastrophic event using the following equation:

$$R = 1 - \frac{2B}{A + B}$$

(2)

Hence, R varied between 1 (no change in the value of $B_s$) and 0 (maximum change) (Fig 3a). In contrast, NC corresponded to the percentage difference between the value of the variable at the end of the simulation ($E$, at $t = 200$) and its value before the catastrophic event ($B$, at $t = 90$), relative to its value before the event (Fig 3b, equation 3).

$$NC = \frac{E - B}{B}$$

(3)

*NC* was therefore negative if $E$ at $t = 200$ was lower than its pre-disturbance value $A$ and positive if it exceeded $A$. Lastly, RR corresponded to the inverse of the recovery time (in years) that $B_s$ took to reach its pre-catastrophic event value, which can also be interpreted as the percentage of mature biomass recovered every year (Fig 3c). If the variable did not retrieve its pre-disturbance value before the end of the simulation, RR was set at 0.01, corresponding to the inverse of the maximum recovery time possible in our simulations (100 years). As such, an increase in each of these three measures represented an increase in the resilience of the mature biomass in the landscape.

We expected R, NC and RR to complement each other since they measure distinct aspects of resilience. Indeed, R measures the magnitude of the initial impact of the catastrophic event but does not consider the temporal dynamic of BS following the event. In contrast, NC and RR are both influenced by the legs of the disturbance event and other sources of mortality (e.g., regular fires) that may occur during the 100-year period of recovery. But while NC revealed how well a stand had recovered during that period, RR showed how fast it had recovered.

In the end, we simply compared the temporal trends and average values for each measure ($B_L$, $\overline{FD}_S$, $B_{FG}$, R, NC and RR) between the scenarios to infer the effects of our different factors. In addition, we use the variability observed between replicates as an indication of uncertainty steaming from the stochastic processes inside our model. We did not use any statistical tests, as the scenario were known to have distinct factors a priori, meaning that any increase in the number of replicates would end up making any difference between scenarios statistically significant [104].

## 3. Results

We present the temporal dynamic of the total mature biomass (BL) and the mean functional response diversity of the forest stands ($\overline{FD}_S$) for each combination of climate scenario, catastrophic disturbance event, and forest management strategy. We also show the total biomass of each functional group (BFG) for scenarios without a catastrophic disturbance event. Additionally, we use bar plots to display how each resilience measure for the mature biomass in the landscape ($B_L$) (resistance R, net change NC and rate of recovery RR) varies across scenarios and replicates following a catastrophe.

### 3.1. Temporal dynamics of mature biomass and mean functional response diversity

Our results show that the temporal dynamic of $B_L$ differed little between the four forest management strategies implemented (Fig 4). In most catastrophe and climate scenarios, the BAU-PlantFunct management strategy tended to increase $B_L$ slightly (by up to 8%) compared the other strategies (Fig 4, yellow curves). In contrast, the normal TRIAD strategy tended to produce smaller $B_L$ (Fig 4, red curves). Furthermore, the variation in $B_L$ between replicates was negligible for each scenario, indicating that the stochasticity of the forest dynamics had little effect when considering the mature biomass at the landscape scale.

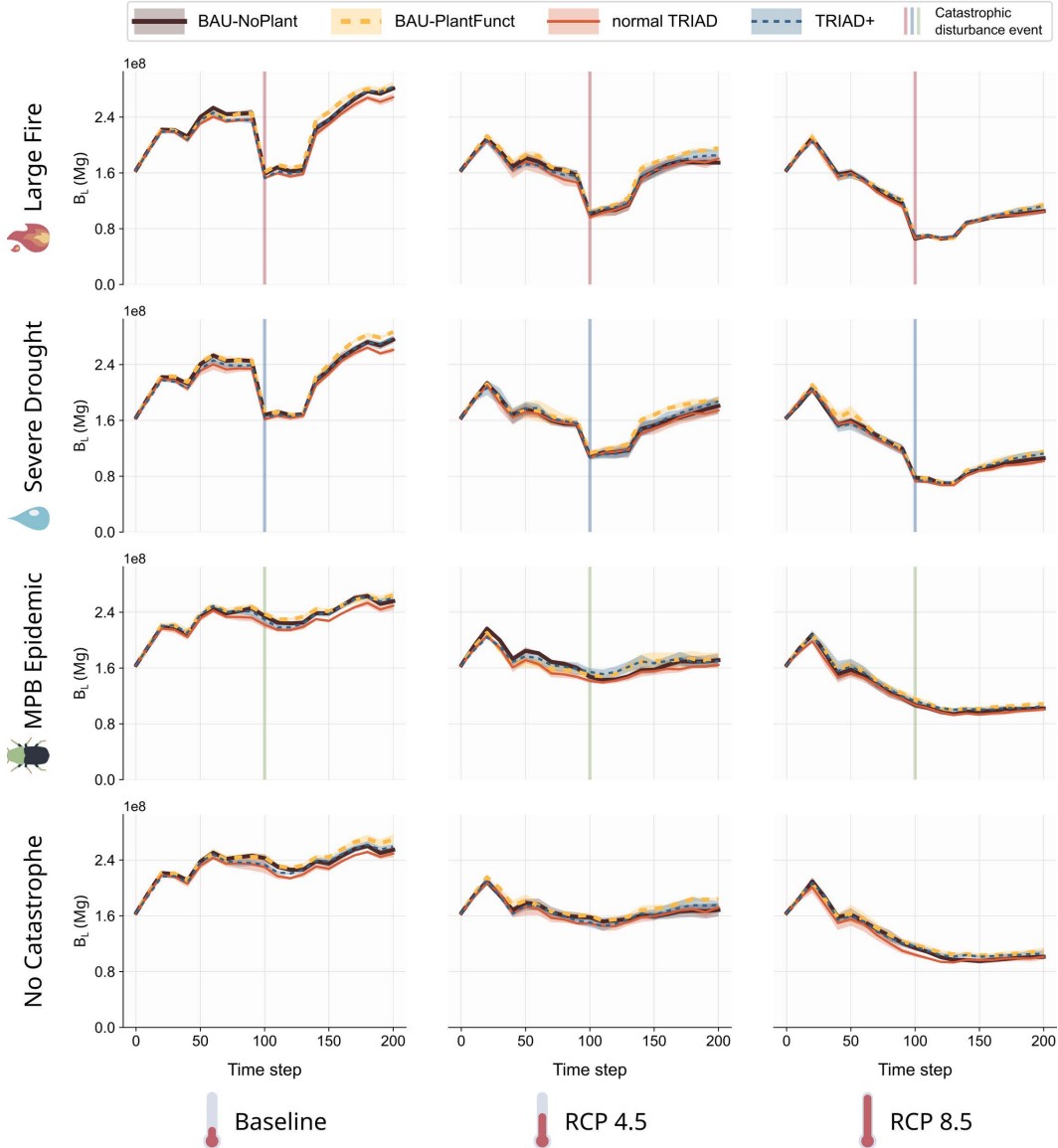

**Fig 4. Temporal variation of the Total Mature Biomass in the landscape $B_L$ for each combination of management, climate and catastrophe scenario.** Solid lines are mean values and envelops are standard deviation across 5 simulation replicates.

The main differences in $B_L$ were observed between the catastrophic disturbance scenarios and between the climate scenarios (Fig 4). Indeed, the large fire and severe drought catastrophes resulted in important reductions in $B_L$ at $t = 100$ (around 30%), impacting its dynamics for the following decades (Fig 4, first and second row). However, $B_L$ ultimately recovered in both cases after 50 years or so and reached values similar to the "no catastrophe" scenarios during the last 50 years of the simulations. In contrast, the MPB outbreak had almost no impact on $B_L$ when compared to the "no catastrophe" scenario (Fig 4, third row). The most important factor impacting the dynamic of $B_L$ was climate. Indeed, $B_L$ either increased in the baseline climate scenario (Fig 4, left column), remained relatively stable in the RCP 4.5 scenario (Fig 4, middle column) or decreased in the RCP 8.5 scenario (Fig 4, right column) throughout the 200 years of the simulations. These trends remained regardless of the forest management strategy involved or nature of the simulated catastrophe.

In contrast, the temporal dynamics of the mean functional diversity, $\overline{FD}S$, varied according to the different forest management strategies (Fig 5). Indeed, regardless of the climate and catastrophe scenario, $\overline{FD}S$ increased by up to 15% in management scenarios with functional planting (TRIAD+ and BAU-PlantFunct, Fig 5, blue and yellow curves) compared to scenarios without functional planting (TRIAD and BAU-NoPlant, Fig 5, red and brown curves). This boost in $\overline{FD}_S$ increased with time but seemed to plateau towards the end of the simulations. However, the wide variability envelopes indicate important variations in functional response diversity between stands and between replicates (Fig 5).

The presence of catastrophes had a small effect on the temporal dynamics of $\overline{FD}S$, in contrast to the results for the total mature biomass (Fig 5, first to last row). However, climate once again played an important role in shaping the

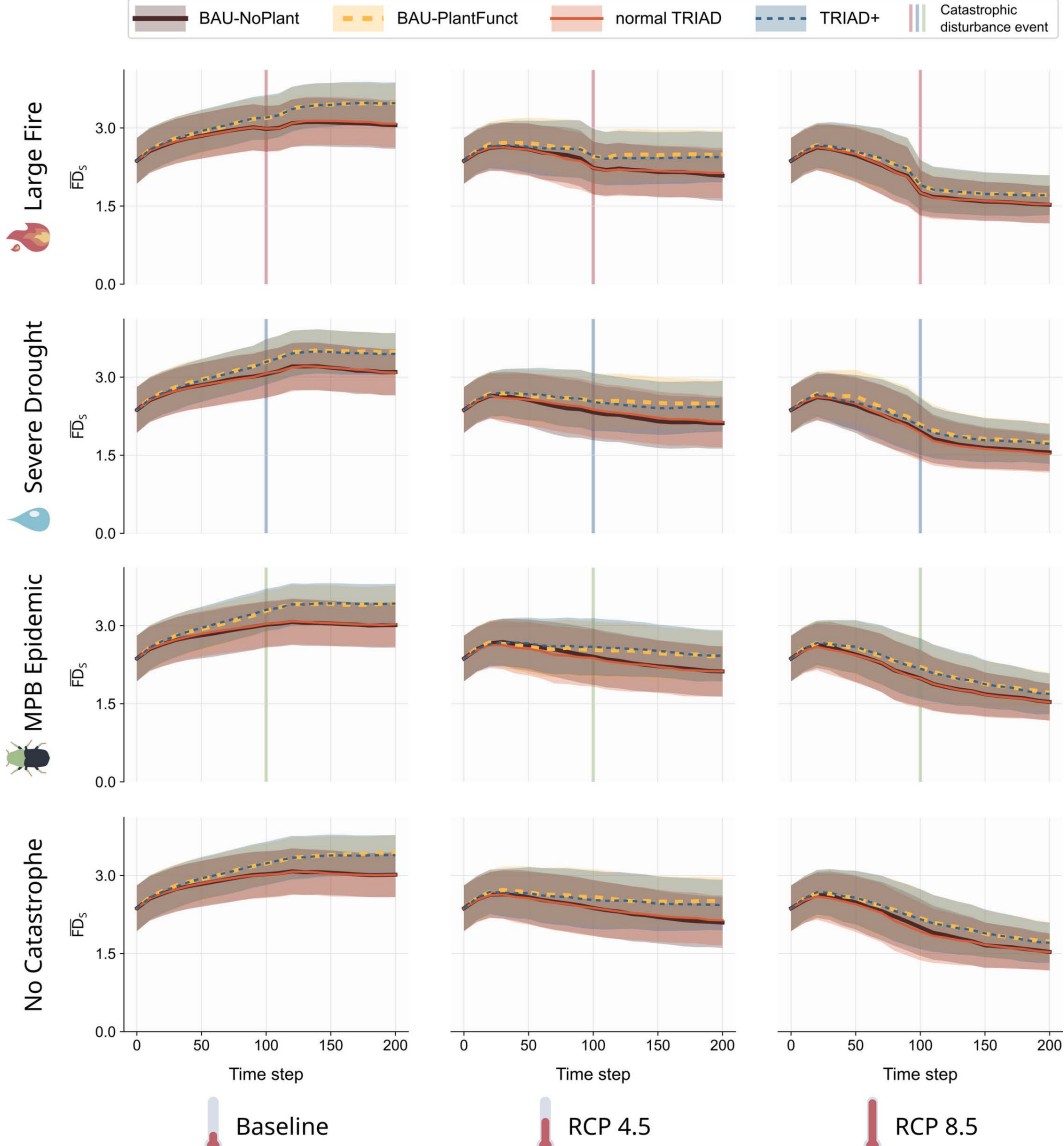

**Fig 5. Temporal variation of the mean Functional Diversity across all stands of the landscape ($\overline{FD}_S$) for each combination of management, climate and catastrophe scenario.** Solid lines are mean values and envelops are standard deviation across stands and 5 simulation replicates.

long-term trend in $\overline{FD}_S$. Indeed, similarly to BL, $\overline{FD}_S$ increased in the baseline scenario (Fig 5, left column), decreased slightly in the RCP 4.5 scenario (Fig 5, middle column), and decreased more steeply in the RCP 8.5 scenario (Fig 5, right column) throughout the simulations.

Finally, the dynamic of total biomass of each functional group ($B_{FG}$) showed very little variation between management scenarios (Fig 6). TRIAD+ and BAU-PlantFunct scenarios show a very slight increase in the biomass of the rarest functional group in the landscape (group 2, softwoods with low tolerance to shade and fire) compared to the TRIAD and BAU-NoPlant scenarios, with a baseline and RCP 4.5 climate (Fig 6, orange areas, first and second row). Overall, the evolution of the biomass of each functional group thus changed according to the climate, but not the management strategy used.

### 3.2. Resilience of mature biomass

Our results show that all resilience measures for $B_L$ were again more sensitive to the simulated climate and catastrophic disturbance event than to the forest management strategy (Fig 7). Nonetheless, in most cases, the mean values of R, NC and RR were higher for the BAU-PlantFunct and TRIAD+ scenarios, pointing to a slightly higher resilience when functional planting was used (Fig 7, yellow and blue bars).

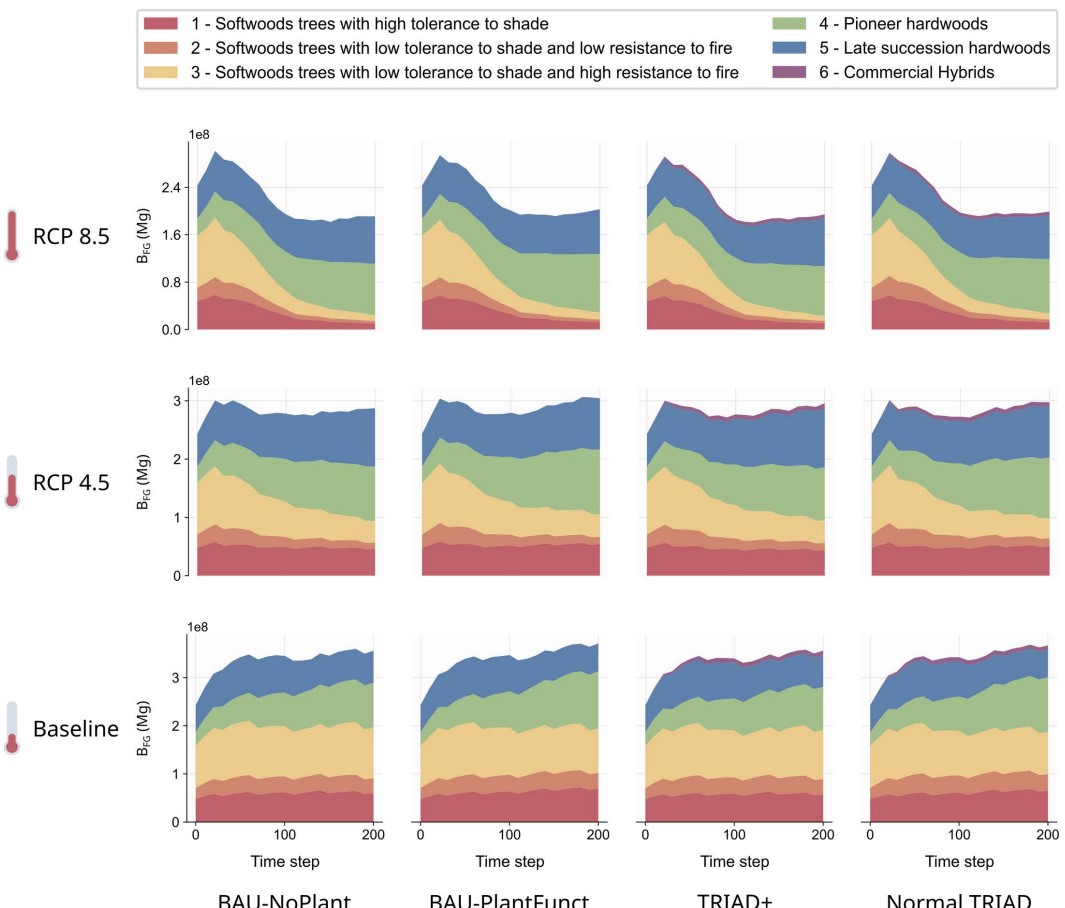

**Fig 6. Evolution of the total biomass for the six functional groups of trees defined in our study for each combination of climate and management scenario, but without a catastrophic disturbance event at t =100.** Values at each time step are mean values across 5 simulation replicates.

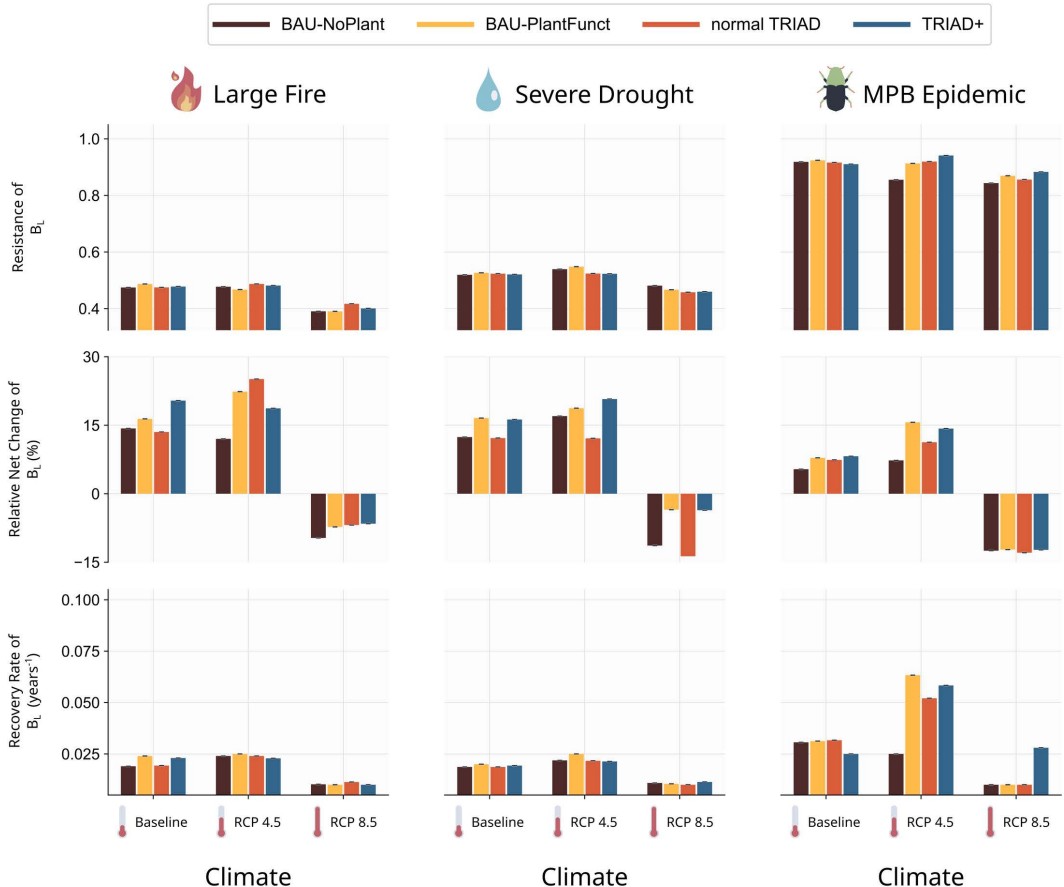

**Fig 7. Bar plots showing the resilience values of the total biomass of the landscape (BL) for each resilience measure (row) and each catastrophic event (column).** The height of the bar are mean values and error lines are standard deviation across 5 simulation replicates.

The resilience measures for $B_L$ varied according to the type of catastrophic disturbance event simulated. The large fire resulted in the smallest values of resistance, R, followed by the severe drought and the MPB outbreak (Fig 7, first row). While the large fire and severe drought resulted in lower values of R than the MPB outbreak (Fig 7, first row), the latter was associated with lower values of NC (Fig 7, middle row). Although $B_L$ was less impacted by the MPB outbreak (leading to higher R values), the landscape did not recover its pre-disturbance $B_L$ as well as with the other catastrophes. On the other hand, the rate of recovery, RR, was slightly higher for the MPB outbreak scenario than for the other two catastrophes, implying a faster recovery of $B_L$ following this outbreak (Fig 7, last row).

Overall, the most coherent signal throughout all resilience measures came from the climate scenarios. Scenarios with the most intense level of climate change (RCP 8.5) was systematically associated with low R, NC and RR values, indicating an overall loss of mature biomass resilience at the landscape scale (Fig 7). However, the RCP 4.5 scenario led to resilience values similar or higher than those for the Baseline scenario for all three measures and all three catastrophes.

## 4. Discussion

Our study aimed at investigating how new forest management strategies could help prepare forests for an uncertain future by increasing their resilience to different extreme disturbance events. To do so, we designed a new type of TRIAD zoning, "TRIAD+", that included functional enrichment planting with the goal of achieving a good compromise between

conservation, production and adaptation in a forest landscape. Our results indicate that the TRIAD+ strategy improved the functional response diversity of forests compared to other management strategies, while increasing the size of protected areas and harvesting the same amount of wood. Our results also show that TRIAD+ increased the resilience of mature biomass in the landscape following different severe disturbances. This could in turn help sustain several important forest ecosystem functions (e.g., seed production, carbon storage, etc.) under unpredictable future extreme events. However, our results also highlight important issues in the practical aspects of functional enrichment: the difficulty of trying to prepare forests for multiple possible extreme events as well as climate changes, and the limitations of landscape-scale models like LANDIS-II to model the effects of functional response diversity.

### 4.1. Climate as the main driver of the dynamics and resilience of the landscape

Among the three factors that we varied in our scenarios, climate stood out as having the greatest effect throughout our simulations (see "Results"). This is particularly apparent when comparing the temporal dynamics of the total mature biomass in the landscape ($B_L$) in scenarios with or without a catastrophe (Fig 4 and 6). In the scenarios with a large fire and severe drought, $B_L$ dropped drastically at t = 100 but rapidly (within 50 years) recovered to similar values found in the corresponding scenarios without catastrophes. In addition, the scenarios without any catastrophe clearly illustrate that $B_L$ decreased with increasing intensity in climate change (Fig 4, bottom row). Taken together, these observations suggest that climate conditions were the strongest drivers determining the dynamics of $B_L$, and not the simulated management strategies nor the punctual disturbances.

Our resilience measures also captured the impact of climate on species' growth. Indeed, scenarios with more intense climate change (i.e., RCP 8.5) presented lower resilience values compared to the baseline and RCP 4.5 scenarios (Fig 7). In the RCP 4.5 scenarios, climate change increased the number of fires (see S1 Appendix E in S1 File) and reduced the growth capabilities of certain species compared to the baseline; however, it also improved the growth of other (thermophilic) species, reducing the differences with baseline scenarios. In contrast, in the RCP 8.5 scenarios, climate increased fire activity and was associated with reduced growth parameters. Our results thus reflect the reported uncertainty regarding the general effect of climate change on forest growth since it may be beneficial through certain processes (e.g., longer growing seasons, CO2 fertilization, etc.; not all considered in this study), but detrimental through others (e.g., more frequent natural disturbances, more hydric stress, etc.) [105,106].

Besides changes in $B_L$, climate change was also responsible for important shifts in $B_{FG}$ (Fig 6). Specifically, the total biomass of functional groups of hardwood species like red oak, trembling aspen, red maple or sugar maple strongly increased throughout all simulations, but even more so under the RCP 4.5 and 8.5 climate scenarios (Fig 6, blue and green areas). On the other hand, the total biomass of functional groups of softwood species like black spruce, white spruce or balsam fir decreased markedly over time, especially under the RCP 4.5 and RCP 8.5 climate scenarios (Fig 6, yellow, orange and red areas). This decrease in softwood biomass resulted in the collapse of the current main economic tree species in Quebec, echoing previous concerns in the literature as to the sustainability of Canada's forestry sector under climate change [107].

### 4.2. Species growth limiting the effect of functional enrichment on functional response diversity

Functional planting, implemented in the TRIAD+ and BAU-PlantFunct scenarios, led to an increase in the mean functional diversity ($\overline{FD}_S$) over time compared to the BAU and TRIAD scenarios (Fig 5). However, this increase plateaued with time – a trend that can be observed for both management scenarios in the temporal evolution of $\overline{FD}_S$ (Fig 5) and of $B_{FG,}$ the total biomass of functional groups (Fig 6). This saturation suggests that fine scale mechanisms, such as species growth and competition, may limit the ability of functional enrichment to diversify this landscape in the long term, especially in the context of climate change. In our model, we implemented functional enrichment by systematically planting a single age cohort of tree species from rare or missing functional groups at the stand scale. Amongst the pool of potential species,

we always selected the species with the highest growth performance relative to the local soil and climatic conditions. However, this planting followed a clear cut that kept 10% of the stand biomass, hence preserving species from the original functional groups. As such, once planted, rarer functional groups would often be unable to compete as their growth parameters (maxANPP) were relatively smaller than those of the original species that were thriving under local climate and soil conditions. This was often the case even though our algorithm selected the most adapted species of the rare functional groups to plant in the given cell. As such, these original species would rapidly outgrow the newly introduced ones. Given that we used a measure of functional diversity weighted by the abundance of functional groups, the increase in $FD_S$ was thus saturated by the limited performance of species from additional functional groups.

### 4.3. Low effect of the functional response diversity on stand resilience

The relatively small effect of our management strategies on the resilience of the mature biomass in the landscape ($B_L$; Fig 7) can be interpreted in two ways. On the one hand, our implementation of functional planting might not have increased the functional response diversity of stands enough to influence their resilience sufficiently (see previous section). On the other hand, the effect of functional response diversity on stand resilience might be small to begin with. Indeed, several empirical studies have observed that the effect of functional response diversity on forest resilience was either absent, small, or highly contextual [108–111]. More generally, Yang et al. [112] argue that functional traits can be poor predictors of tree demographics when the focus is on species rather than individuals, when contextual information about the trait values is missing (e.g., biogeographic, phenotypic), or when functions important to tree demographics are not being captured by the most measured traits.

In contrast to these empirical studies, simulations of tropical forest dynamics using stand-scale individual-based models revealed a positive effect of functional response diversity on forest resilience [113,114]. However, these effects were conditional on how vegetation dynamics was represented in the models. In particular, Schmitt et al. [114] indicated that functional response diversity improved stand resilience to disturbances through complementarity mechanisms between species with different traits (niche partitioning and facilitation). They also noted that the effect of functional response diversity was only temporary as complementarity effects quickly gave way to interspecific competition. Therefore, the effect of functional response diversity on forest resilience may depend on the studied disturbance, the local environmental conditions (slope, etc.) or the forest age. It is also possible that individual-based models are better suited to capture the effect of functional response diversity compared with models, such as LANDIS-II, that are based on age-cohort dynamics (see ""Limitations on model limitations).

In addition, functional enrichment planting with the wide goal of increasing functional response diversity may sometimes be less effective than other forms of planting to increase forest resilience. For example, several studies have found that planting species adapted to specific disturbances or climate conditions could increase the resilience of forest landscapes following these disturbances [115–117]. We argue that this reflects a trade-off between protecting against a broad range of uncertain future conditions and protecting against specific disturbances or aspects of climate change. While the broader strategy might prove less effective when the threats are well known and defined, it might be more effective when facing the unexpected threats that await forests.

### 4.4. Is TRIAD+ a good compromise?

An important goal of our study was to assess whether TRIAD+ could produce a good trade-off between conservation, production and adaptation in the landscape. At first glance, our results might indicate that TRIAD+ was slightly less effective than the BAU-PlantFunct strategy, as the latter resulted in slightly higher values of BL, $\overline{FD}_S$, R, NC and RR in several cases. Specifically, when compared with values for the TRIAD+, the BAU-PlantFunct scenario led to slightly higher values of BL (Fig 4), similar values of $\overline{FD}_S$ (Fig 5), and an equal or slightly improved resilience for the mature biomass in the landscape (Fig 7). However, the performance of the BAU-PlantFunct scenario must be contrasted with

the fact that TRIAD scenarios contained more than twice the quantity of protected areas (reserves) than the BAU scenarios. Indeed, the TRIAD+ scenario had 24% of its forested surface defined as protected areas compared to 9% for the BAU-PlantFunct scenario, and still resulted in similar values of BL, $\overline{FD}_S$ and resilience. Although the tested management scenarios mostly produced small differences in mature biomass, functional diversity, and resilience (S1 Appendix F in S1 File), TRIAD+ can be considered as an appealing avenue to satisfy multiple management goals including resilience to future changes.

### 4.5. Limitations

Our results also suggest some limitations in our modelling methodology. Firstly, the strong effect of climate on the dynamics of BL, $\overline{FD}_S$ and resilience could be explained by how succession dynamics and biomass accumulation are represented in the LANDIS-II Biomass Succession extension we employed. In this extension, each tree species is associated with a maximum annual net primary productivity (ANPP) and a maximum biomass per age cohort that depend on ecoregion and climate [118]. Thus, climate acts on the mature biomass of age cohorts both as a "hard ceiling" via the maximum biomass parameter, and as an "escalator" via the maximum ANPP. Therefore, following any disturbance event, the mature biomass of remaining age cohorts could quickly recover. As such, the strong effect of climate reduced the potential variations in mature biomass caused by other factors, such as forest management and catastrophic events.

Secondly, the Biomass Succession extension may also be responsible for the relatively poor growth of species from rare functional groups. Indeed, the main drivers of forest dynamics in this extension are relatively simple spatially implicit competition for shade and growth, with growth being mostly influenced by climatic conditions in our study area. More complex succession extensions of LANDIS-II, such as the PnET succession extension [119], might therefore have yielded different results by dynamically simulating other mechanisms that influence species' performance at the local scale such as water availability, spatially explicit self-thinning, nutrient competition, or complementarity effects.

### 4.6. Guidelines and considerations for future studies

Our study highlights important considerations on the potential effects of forestry and climate change on future forest ecosystems which can serve as guiding principles in investigating novel resilience-based management strategies. Firstly, the strong effect of climate on the dynamics of mature biomass and functional diversity, compared to the four tested management strategies, suggests that human efforts to shape future forest composition and resilience under climate change will need to be considerable. It will require frequent interventions carried across large scales to successfully guide forest ecosystems into resilient states and maintain this resilience.

Secondly, our results emphasize the challenges surrounding the practice of functional planting. In particular, species selected for functional plantations may not always be able to thrive in the long-term. In our case, part of our functional planting strategy was to select species from rare functional groups. However, even if these new species have functional traits well adapted to future conditions, they may be poor competitors in these novel communities. In such situations, frequent interventions in stands (e.g., via commercial thinning) across a large spatial scale might be required to balance the relative abundance of different functional groups over time. This implies that sustaining the effects of functional enrichment may be a complex and costly operation in certain forests. Other planting strategies focused on specific disturbances or on species expected to "win" under climate change could be considered, although the uncertainty of future forest disturbances may limit the success of such strategies. As such, while the limitations of our study do not allow us to dismiss functional planting as an effective or practical strategy to enhance forest resilience (see previous section), we believe that our results highlight caveats and difficulties that must be considered in future studies. We also believe that while the potentially beneficial effect of functional planting on forest resilience seems to be validated by our model, its importance and efficacy when compared to alternative strategies still remains to be explored, as our results show a relatively small effect.

Thirdly, the contrast between our findings and those of other studies suggest that the simulation of fine-scale processes of forest dynamics should be improved in forest landscape models such as LANDIS-II. Indeed, the intra-cell dynamics in LANDIS-II remains spatially-implicit and does not simulate some small-scale processes such as nutrient and water competition or complementarity effects. As such, in its current form, the ability of LANDIS-II to capture the regeneration dynamics of forest stands following silvicultural intervention is limited.

Fourthly, the collapse in the abundance of several commercial tree species suggests that the construction industry, in tandem with the forest industry, will have no choice but to adapt their practices to stay viable in the future. Both industries will have to harvest and use trees of a more diverse set of species than what is currently the norm [120].

Fifthly, the model assumed that catastrophic disturbances affected all age cohorts equally. However, natural disturbances are known to affect trees of different ages in different ways. For example, recent studies suggest that older and younger trees can differ in their resistance and resilience to drought, due in part to their root system [121,122]. Fire resistance is also expected to increase with age due to bark thickness [92]. In addition, tree age can be an important factor in predicting tree mortality from some insect outbreaks such as the spruce budworm which causes increased mortality to mature trees [123]. Should we have taken these nuances into account, it is possible that our catastrophic events might have influenced forest resilience differently, by favoring either older or younger tree survival depending on the event.

Finally, our study echoes concern that developing multifunctional forest management strategies is no longer a sufficient goal, as shown by the very similar performance of the classic TRIAD and BAU scenarios regarding forest resilience. But management decisions now can shape tomorrow's forests, and therefore must include objectives of adaptation. As the climate warms and humanity's population and resource consumption continue to increase, the future of forests worldwide becomes more and more uncertain. It is therefore our responsibility to find ways to help forests adapt to a world where life on Earth – from humans to trees – is now facing the unknown of the Anthropocene.

## Supporting information

**S1 File. Appendices giving additional figures and details about the parameterization of LANDIS-II.**
(DOCX)

## Acknowledgments

We want to thank Hervé Jactel for his help with the effect size of stand mixture on insect damage.

## Author contributions

**Conceptualization:** Clément Hardy, Christian Messier, Elise Filotas.

**Data curation:** Clément Hardy, Yan Boulanger, Dominic Cyr.

**Formal analysis:** Clément Hardy.

**Funding acquisition:** Christian Messier, Elise Filotas.

**Investigation:** Clément Hardy, Elise Filotas.

**Methodology:** Clément Hardy, Christian Messier, Elise Filotas.

**Project administration:** Elise Filotas.

**Resources:** Christian Messier, Elise Filotas.

**Software:** Clément Hardy, Yan Boulanger, Dominic Cyr.

**Supervision:** Christian Messier, Elise Filotas.

**Validation:** Clément Hardy.

**Visualization:** Clément Hardy.

**Writing – original draft:** Clément Hardy.

**Writing – review & editing:** Christian Messier, Yan Boulanger, Dominic Cyr, Elise Filotas.

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
