## [Decision Letter · Decision Letter 0]

Dear Dr. Hardy,

Thank you for submitting your manuscript to PLOS ONE. After careful consideration, we feel that it has merit but does not fully meet PLOS ONE’s publication criteria as it currently stands. Therefore, we invite you to submit a revised version of the manuscript that addresses the points raised during the review process.

We look forward to receiving your revised manuscript.

Kind regards,

Zhaoxia Guo

Academic Editor

PLOS ONE

Journal Requirements:

3. Thank you for stating the following financial disclosure: Funding was provided by the Natural Sciences and Engineering Research Council of Canada (NSERC) through a Collaborative Research and Development grant awarded to C. Messier (RDCPJ: 498998-16) with the joint support of Resolute Forest Products, and a Discovery grant awarded to E.Filotas (RGPIN: 2018-06156).

4. In the online submission form, you indicated that your data will be submitted to a repository upon acceptance.  We strongly recommend all authors deposit their data before acceptance, as the process can be lengthy and hold up publication timelines. Please note that, though access restrictions are acceptable now, your entire minimal  dataset will need to be made freely accessible if your manuscript is accepted for publication. This policy applies to all data except where public deposition would breach compliance with the protocol approved by your research ethics board. If you are unable to adhere to our open data policy, please kindly revise your statement to explain your reasoning and we will seek the editor's input on an exemption.

Reviewers' comments:

Reviewer's Responses to Questions

**Comments to the Author**

1. Is the manuscript technically sound, and do the data support the conclusions?

Reviewer #1: Yes

Reviewer #2: Yes

2. Has the statistical analysis been performed appropriately and rigorously?

Reviewer #1: Yes

Reviewer #2: No

3. Have the authors made all data underlying the findings in their manuscript fully available?

Reviewer #1: Yes

Reviewer #2: Yes

4. Is the manuscript presented in an intelligible fashion and written in standard English?

Reviewer #1: Yes

Reviewer #2: Yes

Reviewer #1: Dear authors,

Thank you for providing a thorough and insightful analysis of the TRIAD+ approach within forest management to enhance resilience to climate change and extreme disturbances. This study, utilizing LANDIS-II in the Mauricie region, is both timely and relevant as forest managers face increasing pressures to incorporate resilience alongside conservation and production goals.

Paper is well written, English is very good, I have no major remarks. I suggest the minor revision.

The language is appropriate with no flaws (however, not a native speaker). Please find some minor remarks bellow:

Title: /

Abstract: /

Keywords:

it would make sense to add: management approach, disturbances

Comments:

266 first mention of SBW abbreviation should be written in full and SBW in brackets. A latin name should be added where SBW is first mentioned

291 first mention of ANPP abbreviation

SC and CT in Table 1 can also be written in full name

417 is referencing to Table 1 correct?

Figure 4: slight thinner graph lines of different management scenarios and higher resolution of figure would be better

Figure 7 caption typo: Bart plots -> bar plots

Discussion: you will find some supporting results (regarding effect of PlantFunct and climate effect) in a study by BLATTERT et al. 2024. (Managing European Alpine forests with close-to-nature forestry to improve climate change mitigation and multifunctionality)

Reviewer #2: This manuscript titled "Climate is stronger than you think: Exploring functional planting and TRIAD zoning for increased forest resilience to extreme disturbances" addresses a timely and relevant topic in forest management. The integration of resilience components into the TRIAD zoning approach and its evaluation using the LANDIS-II simulation model under various climate and disturbance scenarios provide a novel and interesting perspective. The writing style is clear, the introduction is detailed and engaging, and the methods, results, and discussion are well-structured and informative. The study's findings contribute valuable insights into adapting forest management strategies to the challenges posed by climate change and extreme disturbances. I have some concerns

Concerns Regarding Statistical Support for Results

One notable concern is the absence of statistical tests to support the observed differences in functional diversity and resilience indicators across scenarios. While the manuscript discusses differences between TRIAD+, classic TRIAD, and BAU scenarios, it is unclear whether these differences are statistically significant or within the range of variability expected under the model framework. Including statistical analyses suitable for simulated data would provide stronger evidence for the conclusions drawn and clarify the practical implications of the observed trends.

Inclusion of Additional Metrics for Forest Resilience

The manuscript evaluates forest resilience using three metrics: resistance, net change, and recovery time of mature biomass. These metrics are appropriate but could be complemented by including functional redundancy, an index derived from functional diversity, to provide a more comprehensive assessment of resilience. Functional redundancy reflects the ability of ecosystems to maintain functions despite species loss and could enhance the interpretation of how functional planting influences forest stability under disturbance. I encourage the authors to explore the potential for including this index in their analysis.

Practical Implications of Functional Planting

The results indicate that functional planting provides only modest increases in forest resilience under the simulated scenarios. Given the limited impact observed, the authors should address the practical feasibility of implementing such management strategies, considering the effort, cost, and potential trade-offs involved. Is the modest increase in resilience sufficient to justify the additional efforts required for functional planting? A discussion on this point, perhaps in comparison to alternative or complementary management strategies, would strengthen the manuscript’s practical relevance.

Overall, this study is a significant contribution to forest management literature, and the methodological approach is commendable. Addressing the above concerns, particularly the inclusion of statistical tests and a deeper discussion on practical feasibility, will enhance the manuscript's scientific rigor and applicability.

**Do you want your identity to be public for this peer review?** For information about this choice, including consent withdrawal, please see our Privacy Policy

Reviewer #1: No

Reviewer #2: No

---

## [Author Response · Author response to Decision Letter 1]

16 Apr 2025

We thank both reviewers and the editor tremendously for taking the time to read our manuscript, and suggest changes and edits to improve it.

We have addressed all comments one by one in the file "Response To Reviewers.docx", to which we will refer for our responses.

Thank you again, and happy new year !

---

## [Decision Letter · Decision Letter 1]

Climate is stronger than you think: Exploring functional planting and TRIAD zoning for increased forest resilience to extreme disturbances

PONE-D-24-22896R1

Dear Dr. Hardy,

We’re pleased to inform you that your manuscript has been judged scientifically suitable for publication and will be formally accepted for publication once it meets all outstanding technical requirements.

Kind regards,

Zhaoxia Guo

Academic Editor

PLOS ONE

---

## [Editor Report · Acceptance letter]

PONE-D-24-22896R1

PLOS ONE

Dear Dr. Hardy,

I'm pleased to inform you that your manuscript has been deemed suitable for publication in PLOS ONE. Congratulations! Your manuscript is now being handed over to our production team.

Kind regards,

on behalf of

Dr. Zhaoxia Guo

Academic Editor

PLOS ONE